# Is Transductive Learning Equivalent to PAC Learning?

**Shaddin Dughmi**                                                                         SHADDIN@USC.EDU
*University of Southern California*

**Yusuf Hakan Kalayci**                                                                     KALAYCI@USC.EDU
*University of Southern California*

**Grayson York**                                                                           AGYORK@USC.EDU
*University of Southern California*

**Editors:** Gautam Kamath and Po-Ling Loh

## Abstract

Much of learning theory is concerned with the design and analysis of probably approximately correct (PAC) learners. The closely related transductive model of learning has recently seen more scrutiny, with its learners often used as precursors to PAC learners. Our goal in this work is to understand and quantify the exact relationship between these two models. First, we observe that modest extensions of existing results show the models to be essentially equivalent for realizable learning for most natural loss functions, up to low order terms in the error and sample complexity. The situation for agnostic learning appears less straightforward, with sample complexities potentially separated by a $\frac{1}{\epsilon}$ factor. This is therefore where our main contributions lie. Our results are two-fold:

1. For agnostic learning with bounded losses (including, for example, multiclass classification), we show that PAC learning reduces to transductive learning at the cost of low-order terms in the error and sample complexity. This is via an adaptation of the reduction of Aden-Ali et al. (2023a) to the agnostic setting.

2. For agnostic binary classification, we show the converse: transductive learning is essentially no more difficult than PAC learning. Together with our first result this implies that the PAC and transductive models are essentially equivalent for agnostic binary classification. This is our most technical result, and involves two key steps: (a) A symmetrization argument on the agnostic one-inclusion graph (OIG) of Long (1998) to derive the worst-case agnostic transductive instance, and (b) expressing the error of the agnostic OIG algorithm for this instance in terms of the empirical Rademacher complexity of the class.

We leave as an intriguing open question whether our second result can be extended beyond binary classification to show the transductive and PAC models equivalent more broadly.

**Keywords:** PAC Learning, Transductive Learning, Agnostic Learning, One Inclusion Graphs

## 1. Introduction

The dominant paradigm in statistical learning theory is the probably approximately correct (PAC) model due to Valiant (1984). On the other hand, the transductive model of learning —first introduced in early works by Vapnik and Chervonenkis (1974) and Vapnik (1982), then further developed by Haussler et al. (1994)— does away with distributional assumptions and evaluates a learner's leave-one-out performance on an adversarially-chosen sample. We use the transductive model as it is conceptualized by Daniely and Shalev-Shwartz (2014) in the realizable setting, as well as its natural extension to the agnostic setting by Asilis et al. (2024). The elegant simplicity of the transductive model has enabled application of combinatorial and graph-theoretic insights to learning, as

evidenced by a rich body of work which employs transductive learners as precursors to PAC learners (e.g. Daniely and Shalev-Shwartz (2014); Brukhim et al. (2022); Aden-Ali et al. (2023a); Asilis et al. (2024)). Indeed, some of this recent literature suggests that transductive and PAC learning might be equivalent in a rich variety of learning settings, with essentially identical sample complexities.

The present paper examines the strength of this equivalence. First, we modestly build on the recent work of Aden-Ali et al. (2023a) to show that both realizable and agnostic PAC learning can be reduced to transductive learning when the loss function is bounded, with improved guarantees in the realizable case for pseudometric losses.

This reduction comes at modest expense: a small number of additional samples determined by the error and confidence parameters, as well as multiplication of the error by a universal constant. This implies a strong sense of equivalence of the two models in the realizable setting where it is folklore, and easy to show, that transductive learning reduces to PAC learning at the cost of a constant factor in the error (see e.g. Asilis et al. (2024)).

Such an essentially loss-less reduction from transductive to PAC learning is not known in the agnostic setting. The best we know of is a reduction of Asilis et al. (2024), which loses a factor of $\frac{1}{\epsilon}$ in agnostic sample complexity where $\epsilon$ is the desired error. Other simple approaches seem to suffer the same fate. It therefore remains plausible that agnostic transductive learning is fundamentally more difficult than its PAC counterpart, by up to this $\frac{1}{\epsilon}$ factor in sample complexity. As our second and more technically-involved contribution, we nonetheless rule out such a separation for binary classification. We do so by explicitly quantifying the performance of the optimal agnostic transductive learner in terms of the VC dimension of the class, by way of its empirical Rademacher complexity. This implies that transductive learning serves as a precursor to an essentially optimal PAC learner in agnostic binary classification, via our reduction. We leave open the intriguing question of whether transductive and PAC learning are essentially equivalent more broadly in the agnostic setting.

## 1.1. Background

The overarching goal of statistical learning theory is to "learn" structure from data in a manner that generalizes to unseen future data. In supervised learning, there is a *data domain* $\mathcal{X}$ and *label set* $\mathcal{Y}$. The learner observes a sequence of labeled data points $S = \{(x_1, y_1), \ldots, (x_n, y_n)\} \in (\mathcal{X} \times \mathcal{Y})^n$, called the *sample* or *training data*, from which it derives a *predictor* $h : \mathcal{X} \to \mathcal{Y}$ which anticipates the label of a future *test* data point. The quality of the prediction is evaluated using a *loss function* $\ell : \mathcal{Y} \times \mathcal{Y} \to \mathbb{R}$, applied to the predicted label and the true (unseen) label. The loss of a predictor is compared, in some aggregate sense, to that of the best predictor in some benchmark class $\mathcal{H} \subseteq \mathcal{Y}^{\mathcal{X}}$ of *hypotheses*. The most common loss function we examine in this work is 0-1 loss, i.e. $\ell(y, y') = 0$ if $y = y'$, and $\ell(y, y') = 1$ otherwise. Alternatively, we may use a pseudometric[1] loss, or a bounded loss where $\ell(y, y') \leq c$ for some universal constant $c$.

Since the inception of learning theory, several different data models and benchmarks have been used to evaluate learners (Valiant, 1984; Haussler, 1992; Blumer et al., 1987; Vapnik and Chervonenkis, 1971; Littlestone, 1987; Bousquet et al., 2021). The most prominent theoretical framework for supervised learning is the PAC (probably approximately correct) model due to Valiant (1984). In

---

1. Recall that a metric is a non-negative function $d$ on pairs satisfying that $d(x, x) = 0$ for all $x$, symmetry ($d(x, y) = d(y, x)$ for all $x, y$), the triangle inequality ($d(x, z) \leq d(x, y) + d(y, z)$ for all $x, y, z$), and positivity ($d(x, y) > 0$ whenever $x \neq y$). A pseudo-metric is allowed to violate positivity, i.e., multiple points of the metric space can be at distance 0 from each other.

addition to the hypothesis class $\mathcal{H} \subseteq \mathcal{Y}^{\mathcal{X}}$, a learning problem in this setting fixes a class of plausible data distributions supported on $\mathcal{X} \times \mathcal{Y}$. An instance of the learning problem is then determined by an (unknown) distribution $\mathcal{D}$ in this class, from which both training and test data are drawn. The performance of a predictor $h$ on data drawn from $\mathcal{D}$ is measured by the distributional error (also called distributional risk), defined as the expected loss on a test point sampled from $\mathcal{D}$:

$$L_{\mathcal{D}}(h) := \mathbb{E}_{(x,y)\sim\mathcal{D}}[\ell(h(x), y)].$$

When all data distributions are plausible, this is the *agnostic setting* of learning. When only distributions that are consistent with some hypothesis are plausible, meaning that there exists $h \in \mathcal{H}$ such that $(x, y) \sim \mathcal{D}$ satisfies $y = h(x)$ with probability 1, this is the *realizable setting*. For functions $\epsilon, \delta : \mathbb{N} \to \mathbb{R}_{\geq 0}$, a learner $A$ is said to be an $(\epsilon, \delta)$-PAC learner if, given training data consisting of $n$ i.i.d. draws from $\mathcal{D}$, with probability at least $1 - \delta(n)$ it outputs a predictor whose expected loss on test data from $\mathcal{D}$ is at most $\epsilon(n)$ in excess of that of the best predictor in $\mathcal{H}$, i.e.,

$$\mathbf{Pr}\left[L_{\mathcal{D}}(A(S)) - \min_{h \in H} L_{\mathcal{D}}(h) > \epsilon\right] \leq \delta.$$

Inversely, the sample complexity $m_{\text{PAC}}(\epsilon, \delta)$ of a PAC learner quantifies the minimum number of samples needed to guarantee excess loss $\epsilon$ with probability $1 - \delta$, for given $\epsilon, \delta \geq 0$. When the sample complexity is bounded for every $\epsilon$ and $\delta$, we say the problem is *PAC learnable*.

Much of the work on PAC learning seeks characterizations of learnability for natural problem classes such as binary classification (e.g. Haussler (1995); Blumer et al. (1989)), multi-class classification (e.g. Littlestone (1988); Natarajan (1989); Daniely and Shalev-Shwartz (2014); Brukhim et al. (2022); Daniely et al. (2011)), regression (e.g. Simon (1996); Bartlett et al. (1996); Daniely et al. (2011); Attias et al. (2023)), and beyond. In addition to such qualitative characterizations, a rich literature has established upper bounds (e.g. David et al. (2016); Haussler (1992); Blumer et al. (1987); Bartlett et al. (1996); Daniely et al. (2011); Attias et al. (2023); Hanneke et al. (2024)) and lower bounds (e.g. Valiant (1984); Simon (1996)) on the sample complexity of PAC learning problems.

In the transductive model of learning there is also a hypothesis class $\mathcal{H}$, but no distribution. An instance of the problem consists of a sample $S \in (\mathcal{X} \times \mathcal{Y})^n$ that is selected by an adversary. Of the $n$ labeled data points in this sample, $n - 1$ are selected uniformly at random as the training set, and the trained predictor is applied to the remaining test point. Let $S_{-i}$ stand for the proper subset of $S$ with data point $i$ is taken out. We measure the error of a learner $A$ in the transductive framework as follows:

$$L_S^{\text{Trans}}(A) := \frac{1}{n} \sum_{i \in [n]} \ell(A(S_{-i})(x_i), y_i).$$

The realizable setting of transductive learning, employed first by Haussler et al. (1994) and further developed and formalized by Rubinstein et al. (2009) and Daniely and Shalev-Shwartz (2014), restricts attention to samples that are consistent with some hypothesis in $\mathcal{H}$. The agnostic setting of transductive learning, as originally described by Long (1998) and later elaborated by Asilis et al. (2024), does away with any restrictions on the sample. In both the realizable and agnostic settings, we say that a transductive learner achieves error $\epsilon = \epsilon(n)$ if its expected loss is at most $\epsilon$ in excess of that incurred by the best fixed predictor in $\mathcal{H}$, i.e.,

$$L_S^{\text{Trans}}(A) - \min_{h \in \mathcal{H}} L_S^{\text{Trans}}(h) \leq \epsilon.$$

As in the PAC setting, the sample complexity $m_{\text{Trans}}(\epsilon)$ is defined inversely.

Similar to the PAC learning, numerous studies have investigated the sample complexity of transductive learning for various domains (e.g. Haussler et al. (1994); Haussler (1995); Kupavskii and Zhivotovskiy (2020); Rubinstein et al. (2006); Asilis et al. (2024); Alon et al. (2022); Bartlett and Long (1998)). Notably, Haussler et al. (1994) introduced the one-inclusion-graph (OIG) algorithm for realizable binary classification, which achieves optimal sample complexity (see also Haussler (1995) and Kupavskii and Zhivotovskiy (2020)). Subsequently, Rubinstein et al. (2006) extended this algorithm to realizable multi-class classification, and Bartlett and Long (1998) extended the algorithm to partial concept learning. Recently, Asilis et al. (2024) extended the OIG approach to agnostic classification, also deriving optimal learners.

Since it demands fine-grained sample-by-sample guarantees, the transductive setting poses additional difficulties beyond those of PAC learning. On the other hand, transductive learning demands only expected error guarantees, which fall short of the high probability guarantees required for PAC learning. Despite these differences, Warmuth (2004) asked whether optimal transductive learners can be used as optimal PAC learners, introducing a new perspective into the study of PAC learning. However, Aden-Ali et al. (2023b) negatively resolved this question, showing that optimal transductive learners might not ensure high probability guarantees out of the box.

Subsequent work by Aden-Ali et al. (2023a) revealed that essentially optimal PAC learners for various realizable learning problems could be derived by aggregating the outputs of multiple instances of a transductive learner, trained on different prefixes of the sample. This approach is inspired by online-to-batch conversion methods. Initial studies by Littlestone (1989) and Haussler et al. (1994) developed algorithms that achieved small expected online error and then converted them into offline learners by aggregating models trained on successive sample prefixes. Later, Wu et al. (2022) strengthened these expected-error bounds to high-probability guarantees via a reverse martingale argument. Building on these ideas, Aden-Ali et al. (2023a) proved that combining transductive learners trained on different prefixes results in PAC learners with optimal sample complexity for various realizable problems. Furthermore, the concept of regret– which evaluates a learner against the best expert retrospectively–in online learning resonates with the objective of agnostic learning. This connection suggests the potential for a transductive-to-PAC conversion even in the agnostic setting, a possibility that remained unverified beyond the realizable case until this work.

On the other hand, a well-known folklore result (also shown in (Asilis et al., 2024)) demonstrates that a realizable PAC learner can be converted into a transductive learner with at most a constant increase in the error rate. These results show that, in realizable supervised learning with a large class of loss functions, PAC learning and transductive learning are essentially equivalent.

The situation is more nuanced in the agnostic setting, which is the focus of the present paper. Simple approaches for reducing from transductive to PAC learning suffer a blowup of $1/\epsilon$ in the sample complexity (see e.g. (Asilis et al., 2024, Proposition 69)), and this is the case even for binary classification. We do not know of any better reductions. Conversely, prior to our work there were no known general-purpose reductions from PAC to transductive learning in the agnostic setting that approximately preserve the sample complexity.

Throughout this paper, we say two sample-complexity expressions $m$ and $m'$ are *essentially equivalent* if they differ by at most an additive polynomial term in the error and/or confidence parameters, as well as pre-multiplication of the error and/or confidence by an absolute constant.

When both $m$ and $m'$ are PAC sample complexities, this can be formally stated as

$$m'\left(c \cdot \epsilon, c \cdot \delta\right) - \text{poly}\left(\frac{1}{\epsilon}, \frac{1}{\delta}\right) \leq m\left(\epsilon, \delta\right) \leq m'\left(\frac{\epsilon}{c}, \frac{\delta}{c}\right) + \text{poly}\left(\frac{1}{\epsilon}, \frac{1}{\delta}\right)$$

where $c$ is an absolute constant. When one or both of $m$ and $m'$ is a transductive sample complexity, we simply omit the associated dependence on $\delta$ as needed. This allows for meaningful comparison of sample complexities across transductive and PAC learning problems. We say a learner's sample complexity is *essentially optimal* if it is essentially equivalent to the optimal sample complexity. We say the transductive and PAC models are *essentially equivalent* for a class of problems if the corresponding transductive and PAC sample complexities are essentially equivalent.

## 1.2. Our Results

This paper tackles the following question in settings where it has remained unanswered.

> *Are the PAC and transductive models essentially equivalent in a rich variety of supervised learning problems?*

Aden-Ali et al. (2023a) developed essentially optimal Probably Approximately Correct (PAC) learners by leveraging transductive learners for various classes of realizable learning problems, including classification, partial concept classification, and regression. Combined with folklore reductions in the opposite direction — e.g. (Asilis et al., 2024, Proposition 32) — their results indicate that transductive and PAC learning are essentially equivalent in the realizable setting for a large class of loss functions. We revisit the key results of Aden-Ali et al. (2023a) and present a unified proof that covers the entire spectrum of supervised learning with (pseudo) metric losses. The following informal theorem summarizes our extension:

**Informal Theorem A** *The PAC and transductive models are essentially equivalent for realizable learning with (pseudo) metric losses.*

This extension broadens the class of loss functions for which realizable PAC and transductive problems are essentially equivalent.[2] Although our contribution to this result is minimal and we claim no novelty, we include detailed proofs in Appendix A for completeness. Our proofs provide a slightly different perspective on the existing analysis, which we believe conceptually simplifies the arguments.

The key contribution of our paper is to explore the relationship between PAC and transductive learning in a more complex setting, agnostic learning, where no assumptions are made about the true underlying data distribution. First, we extend the aforementioned reduction to show that agnostic transductive learners can be converted into agnostic PAC learners, at the cost of a constant factor in the error and a few additional samples. This result implies that an agnostic PAC learning problem is essentially no harder than the corresponding agnostic transductive learning problem. The following informal theorem summarizes this finding:

---

2. Note that our results for the agnostic setting, captured in the upcoming Informal Theorem B, further extend this to bounded loss functions by implication. However, the resulting additive increase in sample complexity scales as $\frac{1}{\epsilon^2}$, dominating the optimal sample complexity for most natural realizable learning problems. Therefore, we do not emphasize that implication here.

**Informal Theorem B** *For learning problems with a bounded loss function, an agnostic transductive learner can be transformed into an agnostic PAC learner with essentially equivalent sample complexity.*

Conversely, we establish a lower bound on the sample complexity of transductive learning for agnostic binary classification which essentially matches the sample complexity of the corresponding PAC learning problem. Our approach is not via reduction, but rather through analyzing the agnostic one-inclusion graph of Asilis et al. (2024). First, we employ a symmetrization argument to show that the worst case agnostic transductive error is attained by the uniform distribution on all possible binary labelings. Second, we re-express this error in terms of the empirical Rademacher complexity of the hypothesis class. Finally, we invoke the known relationship between the Rademacher complexity and the VC dimension. Combined with informal Theorem B, we obtain the following.

**Informal Theorem C** *The transductive and PAC models are essentially equivalent for agnostic binary classification.*

This result paves the way for extending the equivalence between PAC and transductive learning from the realizable setting to the agnostic setting. In light of this, we conjecture that PAC and transductive learning are essentially equivalent for a large class of loss functions in the agnostic setting.

**Conjecture 1** *The PAC and transductive models are essentially equivalent for agnostic learning with most natural label spaces and loss functions (including, most notably, multi-class classification).*

In the remainder of the paper, we formalize and prove Informal Theorems B and C. Specifically, Section 2 focuses on Theorem B, while Section 3 is dedicated to Theorem C.

## 2. Reduction for Agnostic Learning

In this section, we prove one side of the "essential" equivalence between the PAC and transductive frameworks in the agnostic setting by showing that any agnostic transductive learner can be converted into an agnostic PAC learner with minimal additional data. The following theorem is the main result of this section. Throughout this section, we assume that $\epsilon_{\text{Trans}}(n)$ is decreasing and that $A(S)$ is symmetric on $S$, meaning $A(\pi(S)) = A(S)$ for any permutation $\pi$. These assumptions hold essentially without loss of generality: most natural learners enjoy these properties, and moreover there always exists such an optimal learner (whether or not we can efficiently find it) — see Appendix C.1 for more detail.

**Theorem 2** *For any agnostic learning problem defined by sample space $\mathcal{X} \times \mathcal{Y}$ with a bounded loss function $\ell(\cdot, \cdot) \in [0, 1]$ and a hypothesis class $\mathcal{H}$, we have*

$$m_{PAC,\mathcal{H}}^{Ag}(\epsilon, \delta) \leq m_{Trans,\mathcal{H}}^{Ag}(\epsilon/4) + \frac{8}{\epsilon^2} \cdot \log\left(\frac{2}{\epsilon \cdot \delta}\right).$$

*Here, $m_{PAC,\mathcal{H}}^{Ag}$ and $m_{Trans,\mathcal{H}}^{Ag}$ represent the agnostic sample complexity of PAC and transductive learning, respectively, for the given hypothesis class $\mathcal{H}$.*

Our strategy for constructing a PAC learner, applicable to both realizable and agnostic settings, utilizes a transductive learner as a subroutine in a manner similar to Aden-Ali et al. (2023a). This entails executing the transductive learner on a set of $k \in \mathrm{poly}(1/\epsilon, \log(1/\delta))$ distinct (not necessarily disjoint) samples. Specifically, let $S = \{(x_1, y_1), \ldots (x_{n+k}, y_{n+k})\}$ be $n + k$ data points, each independently sampled from a distribution $\mathcal{D}$ over $\mathcal{X} \times \mathcal{Y}$. We define a sequence of sample subsets $S_i = \{(x_1, y_1), \ldots (x_{n+i}, y_{n+i})\}$. Our PAC learners call the transductive learner $A_{\mathrm{Trans}}$ independently with samples $S_0, S_1, \ldots S_{k-1}$, generating predictors $h_0, \ldots h_k$.

In order to obtain a randomized predictor that achieves a small distributional loss with high probability over the data distribution but in expectation over its internal randomness, we can simply select a hypothesis at random and return its prediction on each new data point encountered. The following technical lemma shows that the expected error of a randomized predictor is small with high probability.

**Lemma 3** *Let $A_{Trans}$ be a transductive learner with agnostic error rate $\epsilon_{Ag}(n)$. Then for any $\delta \in (0, 1)$, and a sample $S \sim \mathcal{D}^{n+k}$, with probability $1 - \delta$ over the choices of $S$, predictors $h_{i-1} := A_{Trans}(S_{i-1})$ for $i \in [k]$ satisfy:*

$$\mathbf{Pr}\left[\frac{1}{k}\sum_{i=1}^{k} L_{\mathcal{D}}(h_{i-1}) - \min_{h' \in \mathcal{H}} L_{\mathcal{D}}(h') \geq \epsilon_{Ag}(n) + \sqrt{\frac{32 \cdot \log(2/\delta)}{k}}\right] \leq \delta.$$

*In particular, when $k = \left\lceil \frac{32 \cdot \log(2/\delta)}{\epsilon_{Ag}^2(n)} \right\rceil$, the average distributional error is no more than $2 \cdot \epsilon_{Ag}(n)$ with probability at most $1 - \delta$.*

Alternatively, if the goal is to obtain a deterministic learner that achieves small distributional loss with high probability over the data distribution, we must utilize an aggregation step to combine the predictions of $h_0, \ldots, h_k$. In the realizable case and under the metric loss assumption, we showed that using the generalized median of the predictions—that is, the prediction minimizing the sum of losses to the other predictions—ensures good performance. Since the computation of the average distributional error across predictors in the realizable setting closely mirrors the work of Aden-Ali et al. (2023a), and combining predictions via median is a standard trick, we do not claim any novelty for this portion and have therefore deferred the detailed analysis to Appendix A.[3] However, in the agnostic setting, such aggregation strategies are ineffective because they can magnify absolute error. Therefore, we use a small validation set of size $O\left(\frac{1}{\epsilon^2} \log\left(\frac{1}{\epsilon\delta}\right)\right)$ to identify the best predictor among $h_0, \ldots, h_k$. Algorithm 1 summarizes the reduction in the agnostic setting. [4]

Before we prove our technical lemma, Lemma 3, in the upcoming section, we argue briefly how it implies Theorem 2, demonstrating that Algorithm 1 guarantees a small PAC error. Lemma 3 guarantees that the average, and hence also the minimum, of the agnostic risks of $\{h_0, \ldots, h_{k-1}\}$ is bounded by $2 \cdot \epsilon(n)$ with probability $1 - \frac{\delta}{2}$. We then use a hold-out validation dataset to select some $h_i$ whose agnostic risk is within $\epsilon(n)$ of this bound with probability $1 - \frac{\delta}{2}$. Standard Hoeffding bounds

---

3. That said, some readers may find this section interesting as it offers an alternate perspective on the the techniques of Aden-Ali et al. (2023a).

4. Note that this construction assumes that $\delta$ is known in advance in order to determine the appropriate size of the cross validation set. If $\delta$ is unknown, one can still build a randomized learner by selecting a hypothesis uniformly at random from $h_0, \ldots, h_k$, although this approach only yields a guarantee "in expectation" over the learner's internal randomness.

---

**Algorithm 1** Transforming transductive learners to PAC learners in the agnostic setting

**Data:** $n$ i.i.d. samples $S$ from $\mathcal{D}$.

**Result:** Predictor $\widehat{h}$.

Let $A_{\text{Trans}}$ be a transductive learner with agnostic error rate $\epsilon_{\text{Ag}}(n)$.

Define $S_i := (x_1, y_1), \ldots (x_n, y_n), (x_{n+1}, y_{n+1}), \ldots (x_{n+i}, y_{n+i})$ for each $i \in [k]$.

Let $h_{i-1}$ be the output predictor of $A_{\text{Trans}}(S_{i-1})$ for each $i \in [k]$.

Let $S_{\text{Val}}$ be a hold-out validation set of size $k' = O\left(\frac{\log(k/\delta)}{\epsilon_{\text{Ag}}^2(n)}\right)$.

Estimate the empirical error of each predictor $\{h_{i-1}\}_{i \in [k]}$ by using validation set $S_{\text{Val}}$

Set $\widehat{h}$ as the predictor with minimum validation error among $\{h_{i-1}\}_{i \in [k]}$.

**Output:** Predictor $\widehat{h}$.

---

and the union bound imply that a validation set of size $k' = O(\frac{1}{\epsilon^2(n)} \log(\frac{k}{\delta}))$ suffices. Therefore, in total $k + k' = O\left(\frac{1}{\epsilon^2(n)} \log(\frac{1}{\epsilon(n) \cdot \delta})\right)$ samples suffice, as claimed in Theorem 2. We omit the details of the argument here, as it follows a standard Hoeffding bound argument. Please see Appendix B for the proof details.

### 2.1. Proof of Lemma 3

Before we proceed with the proof, we introduce notation to simplify our discussion of agnostic distributional error and agnostic transductive error. For an agnostic PAC learning problem with distribution $\mathcal{D}$ and hypothesis class $\mathcal{H}$, let $h_{\mathcal{D}}^* \in \text{argmin}_{h \in \mathcal{H}} L_{\mathcal{D}}(h)$ denote the optimal hypothesis in $\mathcal{H}$ for the distribution $\mathcal{D}$. We define the *agnostic distributional error* of a predictor $h$ as

$$L_{\mathcal{D},\mathcal{H}}^{\text{Ag}}(h) := L_{\mathcal{D}}(h) - L_{\mathcal{D}}(h_{\mathcal{D}}^*).$$

Similarly, for an agnostic transductive learning problem with sample $S$ and hypothesis class $\mathcal{H}$, let $h_S^* \in \text{argmin}_{h \in \mathcal{H}} L_S^{\text{Trans}}(h)$ be an optimal hypothesis in $\mathcal{H}$ for the sample $S$. We define the *agnostic transductive error* of a learner $A$ as

$$L_{S,\mathcal{H}}^{\text{Trans,Ag}}(A) := L_S^{\text{Trans}}(A) - L_S^{\text{Trans}}(h_S^*).$$

These notations help us to quantify how much worse a predictor or learning algorithm performs compared to the best possible hypothesis within the class $\mathcal{H}$ for the given distribution or sample.

We now proceed to prove Lemma 3. We first define the following random variable:

$$d_i = \ell(h_{i-1}(x_i), y_i) - \ell(h_{\mathcal{D}}^*(x_i), y_i).$$

Observe that conditioned on samples $S_{i-1}$, $d_i$ becomes an unbiased estimator of the distributional agnostic risk of the predictor $h_{i-1}$. Formally,

$$\begin{aligned}
\mathbb{E}[d_i \mid S_{i-1}] &= \mathbb{E}[\ell(h_{i-1}(x_i), y_i) \mid S_{i-1}] - \mathbb{E}[\ell(h_{\mathcal{D}}^*(x_i), y_i) \mid S_{i-1}] \\
&= L_{\mathcal{D}}(h_{i-1}) - L_{\mathcal{D}}(h_{\mathcal{D}}^*) \\
&= L_{\mathcal{D},\mathcal{H}}^{\text{Ag}}(h_{i-1}).
\end{aligned}$$

The proof of Lemma 3 employs forward and backward martingale arguments, mirroring the approach used in the realizable context. In the forward martingale phase, we aim to show that $\sum_{i=1}^{k} L_{\mathcal{D},\mathcal{H}}^{\mathrm{Ag}}(h_{i-1}) \lesssim \sum_{i=1}^{k} d_i$ with high probability. This is achieved by observing that the sum $\sum_{i=1}^{k} L_{\mathcal{D},\mathcal{H}}^{\mathrm{Ag}}(h_{i-1}) - d_i$ consists of random variables with conditional expected value equal to zero, thus forming a martingale. We can then apply Azuma's martingale concentration inequality to show that this supermartingale stays small with high probability. The following claim provides a formal statement of this argument.

**Claim 4 (Forward Martingale Bound)**

$$\mathbf{Pr}\left[\sum_{i=1}^{k} L_{\mathcal{D},\mathcal{H}}^{\mathrm{Ag}}(h_{i-1}) - \sum_{i=1}^{k} d_i > \sqrt{8k \log\left(\frac{2}{\delta}\right)}\right] \leq \delta/2.$$

In the backward martingale argument, we conclude the proof of our technical lemma by showing that $\sum_{i=1}^{k} d_i \lesssim \epsilon$ with high probability. Examining the samples in reverse order models the process as removing one sample at a time uniformly at random. The learner trained on $S_{i-1}$ predicting on $(x_i, y_i)$ effectively computes the transductive error on $S_i$. Since our agnostic transductive learner satisfies $L_{S_i}^{\mathrm{Trans,Ag}} \leq \epsilon$, and by comparing the transductive errors of the best hypothesis for sample, $h_{S_i}^*$, and $h_{\mathcal{D}}^*$, we conclude that $\mathbb{E}[d_i] \leq \epsilon$. Thus, $\sum_{i=1}^{k} d_i$ forms a supermartingale when viewed backward in time. Applying Azuma's inequality confirms that this sum remains small with high probability. The following claim formalizes this argument.

**Claim 5 (Backward Martingale Bound)**

$$\mathbf{Pr}\left[\sum_{i=1}^{k} d_i > k \cdot \epsilon + \sqrt{8k \log\left(\frac{2}{\delta}\right)}\right] \leq \delta/2.$$

Combining claims 4 and 5 using the union bound, we obtain

$$\mathbf{Pr}\left[\sum_{i=1}^{k} L_{\mathcal{D},\mathcal{H}}^{ag}(h_{i-1}) > k \cdot \epsilon + 2\sqrt{8k \log\left(\frac{2}{\delta}\right)}\right] \leq \delta$$

which completes the proof of Lemma 3. As described earlier, Theorem 2 then follows.

In the remainder of this section, we will prove these two claims to complete the proof of Lemma 3.

**Proof** [Proof of Claim 4] Let $M_i = L_{\mathcal{D},\mathcal{H}}^{\mathrm{Ag}}(h_{i-1}) - d_i$, and $\overline{M}_i = \sum_{j=1}^{i} M_j$. Then, consider the following conditional probability to see that $\overline{M}_i$ forms a martingale.

$$\mathbb{E}_{(x_i,y_i)\sim\mathcal{D}}[\overline{M}_i \mid S_{i-1}] = \mathbb{E}[M_i \mid S_{i-1}] + \mathbb{E}[\overline{M}_{i-1} \mid S_{i-1}]$$

$$= \mathbb{E}[L_{\mathcal{D},\mathcal{H}}^{\mathrm{Ag}}(h_{i-1}) - d_i \mid S_{i-1}] + \overline{M}_{i-1}$$

$$= \overline{M}_{i-1}.$$

Since both $d_i$ and $L_{\mathcal{D},\mathcal{H}}^{\mathrm{Ag}}(h_{i-1})$ are bounded between $[-1, 1]$, $M_i$ lies in $[-2, 2]$ and so $|M_i| \leq 2$. Applying Azuma's inequality yields the desired result. ∎

**Proof** [Proof of Claim 5] We think of constructing our sequence of samples $S_0, \ldots, S_k$ "backwards" by first conditioning on the (unordered) set $S_k = \{(x_i, y_i)\}_{i=1}^{n+k}$, then iteratively peeling off one sample at a time — uniformly at random without replacement — to obtain $S_{k-1}, S_{k-2}, \ldots, S_0 = S$. For $i = k, \ldots, 1$, observe that $(x_i, y_i)$ is a uniformly random element from $S_i$, and that $|S_i| \geq n$. The transductive error assumption implies that

$$L_{S_i, \mathcal{H}}^{\text{Trans,Ag}}(A) = \mathbb{E}\left[\ell(h_{i-1}(x_i), y_i) - \ell(h_{S_i}^*(x_i), y_i)|S_i\right] \leq \epsilon. \tag{1}$$

Similarly, because $h_{S_i}^*$ is the hypothesis minimizing transductive loss for the sample $S_i$, we know that the loss of any other hypothesis on $S_i$ must exceed that of $h_{S_i}^*$. Therefore,

$$L_{S_i}^{\text{Trans}}(h_{S_i}^*) \leq L_{S_i}^{\text{Trans}}(h_{\mathcal{D}}^*)$$

or equivalently saying that

$$\mathbb{E}\left[\ell(h_{S_i}^*(x_i), y_i) - \ell(h_{\mathcal{D}}^*(x_i), y_i)|S_i\right] \leq 0. \tag{2}$$

Here, the expectation is taken over a uniformly random selection of the test point.

Adding 1 and 2, we see that

$$\mathbb{E}\left[\ell(h_{i-1}(x_i), y_i) - \ell(h_{\mathcal{D}}^*(x_i), y_i)|S_i\right] = \mathbb{E}[d_i|S_i] \leq \epsilon.$$

Next, we define random variables $B_i = d_i - \epsilon$, $\overline{B}_i = \sum_{j=i}^{k} B_i$ and observe that

$$\mathbb{E}\left[\overline{B}_i \mid \overline{B}_{i+1}\right] = \overline{B}_{i+1} + \mathbb{E}\left[d_i - \epsilon \mid d_k, \ldots, d_{i+1}\right] \leq \overline{B}_{i+1}$$

Thus, the sequence $\overline{B}_i$ is a backwards super-martingale — i.e., a super-martingale when viewed backwards in time. Invoking Azuma's inequality, we obtain

$$\mathbf{Pr}\left[\overline{B}_1 > k \cdot \epsilon(n)\right] = \mathbf{Pr}\left[\sum_{i=1}^{k} \widetilde{d}_i > 2 \cdot k \cdot \epsilon(n)\right] \leq \frac{\delta}{2}.$$

∎

While this analysis for the agnostic case resembles that of the realizable case in (Aden-Ali et al., 2023a) and Appendix A, we will mention two key differences. First, a multiplicative tail bound (e.g. the multiplicative version of Azuma's inequality) is effective in the realizable setting because the expected absolute error is close to zero. In contrast, a small agnostic error only ensures the learner's performance is not much worse than the best hypothesis in the class, whereas the algorithm's absolute error can still be large. Using a multiplicative tail bound here would amplify the error and compromise the PAC sample complexity. Therefore, we use the traditional additive version of Azuma's inequality to control the error without such amplification. Second, in the realizable setting the learner does not need to be compared against a benchmark hypothesis. In contrast, in the agnostic setting, the performance of the predictor must be compared to the best hypothesis in the class when evaluated on the distribution and on the transductive sample respectively. Therefore, in this argument, we must ensure that the difference in these benchmark terms does not introduce significant additional error.

## 3. Optimal Agnostic Transductive Learner for Binary Classification

In this section, we aim to establish the agnostic transductive error rate of binary classification in terms of the VC dimension of the benchmark hypothesis class $\mathcal{H}$. We analyze the agnostic OIG algorithm of Asilis et al. (2024) and show that together with our reduction, the agnostic OIG algorithm obtains essentially optimal PAC sample complexity. This result indicates that the PAC and transductive learning frameworks are essentially equivalent for binary classification with 0-1 loss, thereby confirming Informal Theorem C. The main result of this section is presented in the following theorem:

**Theorem 6** *The Agnostic One Inclusion Graph algorithm (Asilis et al., 2024) for transductive binary classification with benchmark hypothesis class $\mathcal{H} \subseteq \mathcal{Y}^{\mathcal{X}}$ attains agnostic error rate of $\epsilon(n) = 16 \cdot \sqrt{\frac{\mathrm{VC}(\mathcal{H})}{n}}$ and sample complexity of at most $O\left(\frac{\mathrm{VC}(\mathcal{H})}{\epsilon^2}\right)$.*

Let $A_{\mathrm{Trans}}$ denote the agnostic transductive learner guaranteed by this theorem. By invoking Theorem 2, we can infer that Algorithm 1 transforms $A_{\mathrm{Trans}}$ into a PAC learner with a sample complexity of $O\left(\frac{\mathrm{VC}(\mathcal{H})+\log(1/(\epsilon\delta))}{\epsilon^2}\right)$. Since the fundamental theorem of statistical learning (Shalev-Shwartz and Ben-David, 2014) provides a lower bound of $\Omega\left(\frac{\mathrm{VC}(\mathcal{H})+\log(1/\delta)}{\epsilon^2}\right)$ on agnostic PAC sample complexity, the error rate in our theorem is optimal up to a constant factor.

In the remainder of this section, we start with an overview of the One Inclusion Graph algorithm and its agnostic variant, followed by a proof of Theorem 6.

### 3.1. Preliminaries of the One Inclusion Graph Algorithm

The One Inclusion Graph (OIG) algorithm was first introduced by Haussler et al. (1994) to design learners for binary classification within the transductive learning framework. This concept was later extended to various other learning problems, such as multi-class classification (Rubinstein et al., 2009) and regression (Attias et al., 2023). However, since this section focuses on binary classification, we will define OIGs specifically for binary classification and explore the relevant results.

Given a sample set $T \in \mathcal{X}^n$, consisting of (unlabeled) datapoints, the one-inclusion graph (OIG) $G_{\mathcal{H}_{|T}} = (V, E)$ is defined as follows. The vertex set $V = \mathcal{H}_{|T}$ consists of restrictions of $\mathcal{H}$ to $T$, i.e., all possible classifications of $T$ by hypothesis class $\mathcal{H}$. The edge set $E$ contains an edge for every pair of restricted hypotheses $h, h'$ if they disagree at exactly one data point from $T$. An orientation of the edges of this graph can immediately be interpreted as a transductive learner, with error proportional to the maximum outdegree. Haussler et al. (1994) showed that if the edge density is low in all subgraphs simultaneously, there exists an orientation with small maximum outdegree, and therefore a transductive learner with small error. In particular, Haussler et al. (1994) showed that when $\mathcal{H}$ has finite VC dimension, the OIG algorithm attains an optimal transductive error rate of $\epsilon(n) = \frac{\mathrm{VC}(\mathcal{H})}{n}$. Later works derived this error rate using different techniques (Haussler et al., 1994; Haussler, 1995; Kupavskii and Zhivotovskiy, 2020).

Recently, Asilis et al. (2024) studied the agnostic-OIG algorithm introduced by Long (1998), where they show it also achieves the optimal error. Consider an agnostic binary classification problem with a benchmark hypothesis class $\mathcal{H} \subseteq \mathcal{Y}^{\mathcal{X}}$, where $\mathcal{Y} = \{+1, -1\}$, and let $T \in \mathcal{X}^n$ be an unlabeled sample. The agnostic one-inclusion graph $G^{\mathrm{Ag}}_{\mathcal{H}_{|T}} = (V^{\mathrm{Ag}}, E^{\mathrm{Ag}})$ is defined as follows:

- $V^{\mathrm{Ag}} = \mathcal{Y}^n$, representing one node for each possible labeling of the $n$ data points.

- $E^{\mathrm{Ag}} = \{(u, v) : \|u - v\|_0 = 1\}$, where the Hamming distance $\|u - v\|_0$ is defined as $\sum_{i=1}^{n} [u_i \neq v_i]$, representing the number of indices at which $u$ and $v$ differ.

Most importantly, each $u \in V^{\mathrm{Ag}}$ is associated with a *credit* equal to its Hamming distance to the realizable classifications of $T$, namely $\|u - \mathcal{H}_{|T}\|_0 = \min_{v \in \mathcal{H}_{|T}} \|u - v\|_0$. It is worth noting that $G^{\mathrm{Ag}}_{\mathcal{H}_{|T}}$ is structurally isomorphic to the Boolean hypercube of dimension $|T| = n$.

The agnostic one inclusion graph algorithm, adapting its realizable counterpart, orients the edges of the agnostic OIG to ensure that each vertex has a small *excess out-degree*, measured relative to its credit. A labeling of $n - 1$ data points (the training data) corresponds to an edge of this agnostic OIG, and the algorithm labels the test point in accordance with the vertex chosen by the oriented edge.

Now, we recall the "discounted edge density" of subgraphs of the agnostic OIG, and restate a lemma from Asilis et al. (2024) which relates the maximum discounted edge density to the error rate of the agnostic OIG algorithm. The *discounted edge density* of a set $U \subseteq V^{\mathrm{Ag}}$ is defined as

$$\Phi_{\mathrm{discounted}}(U) := \frac{|E(U, U)| - \sum_{u \in U} \|u - \mathcal{H}_{|T}\|_0}{|U|}.$$

A familiar reader might recognize that the left term of the ratio in the definition of $\Phi_{\mathrm{discounted}}$ is actually the subgraph edge density introduced by Haussler et al. (1994). In the agnostic setting, the negative term sums the credits of the nodes in $U$, which represent how far each particular class assignment is from the hypothesis class $\mathcal{H}$. All else held equal, larger credit values let the learner "off the hook", leading to lower agnostic transductive error rates. In a sense, credit measures how suboptimal this hypothesis class is for general assignments, thereby allowing a learner to compete with $\mathcal{H}$. Intuitively, an adversary who seeks to maximize the expected error of the learner will choose $U$ maximizing discounted edge density, then sample a labeling from $U$ uniformly at random. This is formalized in the following Lemma from Asilis et al. (2024).

**Lemma 7 (Implied by Asilis et al. (2024))** *The agnostic OIG algorithm attains the optimal agnostic error rate for classification in the transductive model, expressed as*

$$\epsilon_{Ag}(n) = \frac{1}{n} \cdot \max_{T \in \mathcal{X}^n} \max_{U \subseteq V^{Ag}} \Phi_{discounted}(U)$$

*where $V^{Ag}$ denotes the vertex set of $G^{Ag}_{\mathcal{H}_{|T}}$.*

### 3.2. Proof of Theorem 6

In the rest of this section, we use Lemma 7 to prove Theorem 6. Our proof consists of two parts, stated below as technical lemmas.

First, we show that for an arbitrary unlabeled sample set $T$ and its agnostic OIG $G^{\mathrm{Ag}}_{\mathcal{H}_{|T}} = (V^{\mathrm{Ag}}, E^{\mathrm{Ag}})$, the discounted edge density increases when the subset of labelings $U \subseteq V^{\mathrm{Ag}}$ is grown to a set which is symmetric with respect to the labeling of an arbitrary data point. Recall that the label set is $\mathcal{Y} = \{+1, -1\}$. We define two sets of classifiers, $U_{i \to +}$ and $U_{i \to -}$, each containing copies of every classifier $h$ from $U$, with the decision for the $i$-th data point modified to class $+1$

and class $-1$, respectively. The *symmetrization* of $U$ with respect to the $i$-th data point is defined as $U' = U_{i \to +} \cup U_{i \to -}$. We demonstrate that the discounted edge density of $U'$ is always greater than that of $U$. A visual representation of this argument is provided in Figure 1. This observation implies that the discounted edge density is maximized for the complete set of classifiers, $V^{\mathrm{Ag}}$, as formalized in the following theorem:

**Lemma 8** *Given an agnostic binary classification problem, let $G^{Ag}_{\mathcal{H}_{|T}} = (V^{Ag}, E^{Ag})$ be the agnostic OIG corresponding to an unlabeled sample $T$ and a hypothesis class $\mathcal{H}$. The following inequality holds for all $U \subseteq V^{Ag}$:*

$$\Phi_{discounted}(U) \leq \Phi_{discounted}(V^{Ag}).$$

In the second step, we estimate the discounted edge density of the complete set of labelings $V^{\mathrm{Ag}}$. Through algebraic manipulations we show that for any sample $T = \{x_i\}_{i=1}^n$ of size $n$:

$$\Phi_{\mathrm{discounted}}(V^{\mathrm{Ag}}) = O(n) \cdot \widehat{\Re}_T(\mathcal{H}_{|T})$$

where $\widehat{\Re}_T(\mathcal{H}_{|T})$ is the empirical Rademacher complexity of $\mathcal{H}_{|T}$, defined as

$$\widehat{\Re}_T(\mathcal{H}_{|T}) := \frac{1}{n} \mathop{\mathbb{E}}_{\sigma \sim \{+1, -1\}^n} \left[ \sup_{h \in \mathcal{H}_{|T}} \sum_{i=1}^n h(x_i) \cdot \sigma_i \right].$$

By utilizing the well-known upper bound of Rademacher complexity in terms of the VC dimension of $\mathcal{H}$, we conclude the following lemma:

**Lemma 9** *Given an unlabeled sample $T$ and hypothesis class $\mathcal{H}$, let $G^{Ag}_{\mathcal{H}_{|T}} = (V^{Ag}, E^{Ag})$ be their agnostic OIG. Then, we have $\Phi_{discounted}(V^{Ag}) \leq 16 \cdot \sqrt{n \cdot \mathrm{VC}(\mathcal{H})}$.*

Before presenting the proofs of these lemmas, we demonstrate how they collectively establish the main result of this section. For any arbitrary unlabeled sample set $T \in \mathcal{X}^n$ of size $n$, consider its agnostic one-inclusion graph $G^{\mathrm{Ag}}_{\mathcal{H}_{|T}} = (V^{\mathrm{Ag}}, E^{\mathrm{Ag}})$. By Lemmas 8 and 9, we have:

$$\max_{U \subseteq V} \Phi_{\mathrm{discounted}}(U) = \Phi_{\mathrm{discounted}}(V^{\mathrm{Ag}}) \leq 16 \cdot \sqrt{n \cdot \mathrm{VC}(\mathcal{H})}.$$

Since $T$ is arbitrary, invoking Lemma 7 ensures that the agnostic error rate is bounded by $16 \cdot \sqrt{\frac{\mathrm{VC}(\mathcal{H})}{n}}$, thus completing the proof of Theorem 6.

The remainder of this section is devoted to the proofs of Lemmas 8 and 9.

### 3.3. Proof of Lemma 8: Symmetrization Argument

For simplicity, we omit the superscript "Ag" and will refer to $V^{\mathrm{Ag}}$ and $E^{\mathrm{Ag}}$ as $V$ and $E$, respectively, throughout the remainder of the paper. Recall that we define the total credit (or total Hamming distance) as $\|U - \mathcal{H}_{|T}\|_0 := \sum_{u \in U} \|u - \mathcal{H}_{|T}\|_0$ for any subset $U \subseteq V$. Each vertex $v \in V$

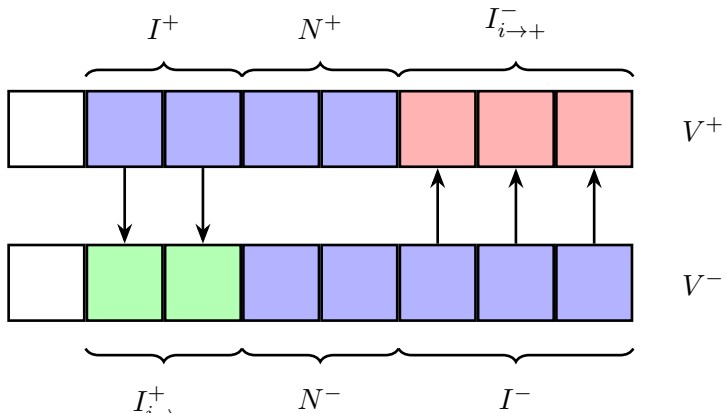

Figure 1: **Symmetrization argument:** A Boolean hypercube of dimension 4 is partitioned into two smaller hypercubes of dimension 3 by fixing a certain coordinate $i$. The smaller hypercubes $V^+$ and $V^-$ are visualized as 8 boxes in the figure. Vertically aligned boxes correspond to bit strings that differ only at index $i$. The set $U$ is indicated by the blue boxes.

represents a labeling for the samples in $T$. We define $v_{i\to+}$ and $v_{i\to-}$ as new vertices where the $i$-th sample is assigned the class label $+1$ and $-1$, respectively. Formally,

$$v_{i\to+}(x_j) = \begin{cases} v(x_j) & \text{for any } x_j \in T_{-i} \\ +1 & j = i. \end{cases} \qquad v_{i\to-}(x_j) = \begin{cases} v(x_j) & \text{for any } x_j \in T_{-i} \\ -1 & j = i. \end{cases}$$

We extend this definition to subsets of vertices as $U_{i\to+} := \{u_{i\to+} : u \in U\}$ and $U_{i\to-} := \{u_{i\to-} : u \in U\}$.

Our aim is to show that $\Phi_{\text{discounted}}(V) \geq \Phi_{\text{discounted}}(U)$ for any $U \subseteq V$. Let $\alpha := \max_{U \subseteq V} \Phi_{\text{discounted}}(U)$ and let $U$ be an arbitrary non-empty subset satisfying $\Phi_{\text{discounted}}(U) = \alpha$. Without loss of generality, we can assume that $\alpha > 0$ and $U$ is non-empty. Otherwise, $\Phi_{\text{discounted}}(U)$ becomes zero uniformly for all subsets which trivially implies the lemma.

Next, we define

$$g(U') := |E(U', U')| - \|U' - \mathcal{H}_{|T}\|_0 - \alpha \cdot |U'|$$

for any $U' \subseteq V$. Recall that $E(U, U)$ is the collection of edges with both endpoints in $U$.

Next, we aim to demonstrate that $g(V) \geq 0$. If this condition is satisfied, it follows that $\Phi_{\text{discounted}}(V) \geq \alpha$, thereby completing the proof.

Fix an arbitrary index $i \in [n]$. Let $V^+$ and $V^-$ form a partition of $V$ such that $V^+ := \{v \in V : v(x_i) = +1\}$ and $V^- := \{v \in V : v(x_i) = -1\}$. By using the sets $V^+$ and $V^-$, we also define a partition of $U$ as $U^+ = U \cap V^+$ and $U^- = U \cap V^-$. We further define the following sets:

$$N^+ = U^+ \cap U^-_{i\to+} \qquad\qquad N^- = U^- \cap U^+_{i\to-}$$
$$I^+ = U^+ \setminus N^+ \qquad\qquad\qquad I^- = U^- \setminus N^-.$$

Here, $N^+, I^+, N^-, I^-$ form a partition of $U$. The intuition behind this partitioning is to analyze the behavior of the function $g$ when the $i$-th bit is flipped for each part of the set $U$. We refer the reader to Figure 1 for a visualization of these sets.

We define $U' = U_{i\to+} \cup U_{i\to-}$, i.e., $U'$ is a superset of $U$ which is symmetric with respect to $i$-th coordinate. In other words, for any vertex $v \in U$, $v_{\to+}, v_{\to-} \in U'$. Next, we will show that $g$ increases as we make the set $U$ symmetric at any coordinate.

**Claim 10** *For all $U \subseteq V$, $g(U') \geq g(U)$, where $U' = U_{i\to+} \cup U_{i\to-}$*

**Proof** We start by observing that

$$g(U) = g(I^+) + g(I^-) + g(N^+ \cup N^-) + |E(I^+, N^+)| + |E(I^-, N^-)|.$$

Similarly, for $U'$ we evaluate $g(U')$ as follows:

$$\begin{aligned} g(U') = g(U) &+ g(I^+_{i\to-}) + g(I^-_{i\to+}) \\ &+ |E(I^+_{i\to-}, U^-)| + |E(I^-_{i\to+}, U^+)| \\ &+ |E(I^+_{i\to-}, I^+)| + |E(I^-_{i\to+}, I^-)|. \end{aligned} \quad (3)$$

Now, we make series of observations:

1. For each vertex $v \in V$, because $\|v_{i\to-} - v_{i\to+}\|_0 = 1$ we see by the triangle inequality that

   $$\max\{\|v_{i\to+} - \mathcal{H}_{|T}\|_0, \|v_{i\to-} - \mathcal{H}_{|T}\|_0\} \leq \min\{\|v_{i\to+} - \mathcal{H}_{|T}\|_0, \|v_{i\to-} - \mathcal{H}_{|T}\|_0\} + 1.$$

   Therefore, we have

   $$\|v_{i\to+} - \mathcal{H}_{|T}\|_0 + \|v_{i\to-} - \mathcal{H}_{|T}\|_0 \leq 2 \cdot \min\{\|v_{i\to+} - \mathcal{H}_{|T}\|_0, \|v_{i\to-} - \mathcal{H}_{|T}\|_0\} + 1.$$

2. From the first observation we deduce that

   $$\|I^+_{i\to-} - \mathcal{H}_{|T}\|_0 \leq \|I^+ - \mathcal{H}_{|T}\|_0 + |I^+| \qquad \text{and} \qquad \|I^-_{i\to+} - \mathcal{H}_{|T}\|_0 \leq \|I^- - \mathcal{H}_{|T}\|_0 + |I^-|.$$

   Since $|E(I^+_{i\to-}, I^+_{i\to-})| = |E(I^+, I^+)|$ and $|E(I^-_{i\to+}, I^-_{i\to+})| = |E(I^-, I^-)|$, subtracting these quantities from the previous inequalities, we see that

   $$g(I^+_{i\to-}) \geq g(I^+) - |I^+| \qquad \text{and} \qquad g(I^-_{i\to+}) \geq g(I^-) - |I^-|$$

3. For edge sets we observe

   $$|E(I^+_{i\to-}, U^-)| \geq |E(I^+, N^+)| \qquad \text{and} \qquad |E(I^-_{i\to+}, U^+)| \geq |E(I^-, N^-)|$$

   and also

   $$|E(I^+_{i\to-}, I^+)| = |I^+| \qquad \text{and} \qquad |E(I^-_{i\to+}, I^-)| = |I^-|.$$

By substituting these inequalities into (3) we obtain that

$$
\begin{aligned}
g(U') - g(U) &\geq g(I^+) - |I^+| + g(I^-) - |I^-| + |E(I^+, N^+)| + |E(I^-, N^-)| + |I^+| + |I^-| \\
&= g(I^+) + g(I^-) + |E(I^+, N^+)| + |E(I^-, N^-)| \\
&= g(U) - g(N^+ \cup N^-).
\end{aligned}
$$

We claim that $g(U) - g(N^+ \cup N^-)$ is non-negative, since assuming otherwise implies that $g(N^+ \cup N^-) > 0$ and therefore $\Phi_{\text{discounted}}(N^+ \cup N^-) > \alpha$, contradicting the optimality of $U$. Thus, $g(U') \geq g(U)$, and consequently $\Phi_{\text{discounted}}(U') \geq \Phi_{\text{discounted}}(U)$. ∎

By construction, the set $U'$ is the minimal superset of $U$ that is symmetric with respect to the index $i$. Iteratively applying this process for each index $i \in [n]$ will always weakly increase the value of $g$ and, consequently, $\Phi_{\text{discounted}}$. Thus, the maximal subset $U^*$ that satisfies $g(U^*) \geq 0$, or equivalently $\Phi_{\text{discounted}}(U^*) = \alpha$, is symmetric with respect to any index. The only non-empty subset with this property is $V$ itself, i.e., the complete Boolean hypercube. Therefore, $\Phi_{\text{discounted}}(V) \geq \Phi_{\text{discounted}}(U)$, completing the proof.

### 3.4. Proof of Lemma 9: Connections to Rademacher Complexity

Given an arbitrary set of samples $T \in \mathcal{X}^n$ we write $\Phi_{\text{discounted}}(V)$ as follows.

$$
\begin{aligned}
\Phi_{\text{discounted}}(V) &= \frac{|E(V, V)| - \sum_{u \in V} \|u - \mathcal{H}_{|T}\|_0}{|V|} \\
&= \frac{|E(V, V)| - \sum_{u \in V} \min_{h \in \mathcal{H}_{|T}} \|u - h\|_0}{|V|} && \text{(definition of } \|u - \mathcal{H}_{|T}\|_0) \\
&= \frac{n \cdot 2^{n-1} - \sum_{u \in V} \min_{h \in \mathcal{H}_{|T}} \|u - h\|_0}{2^n} && (|E| = n \cdot 2^{n-1}, |V| = 2^n) \\
&= \frac{n}{2} - \frac{1}{2^n} \sum_{u \in V} \min_{h \in \mathcal{H}_{|T}} \|u - h\|_0.
\end{aligned}
$$

Next, we switch notation from Hamming distance to inner product by using the following equality.

$$
\langle u, h \rangle := \sum_{i=1}^{n} u(x_i) \cdot h(x_i) = \sum_{i=1}^{n} [u(x_i) = h(x_i)] - \sum_{i=1}^{n} [u(x_i) \neq h(x_i)] = n - 2 \cdot \|u - h\|_0. \quad (4)
$$

Then,

$$
\begin{aligned}
\Phi_{\text{discounted}}(V) &= \frac{n}{2} - \frac{1}{2^n} \sum_{u \in V} \min_{h \in \mathcal{H}_{|T}} \frac{n - \langle u, h \rangle}{2} && (4) \\
&= \frac{n}{2} - \frac{n}{2} - \frac{1}{2^n} \sum_{u \in V} \min_{h \in \mathcal{H}_{|T}} -\frac{\langle u, h \rangle}{2} && (|V| = 2^n) \\
&= \frac{1}{2} \cdot \frac{1}{2^n} \sum_{u \in V} \max_{h \in \mathcal{H}_{|T}} \langle u, h \rangle \\
&= \frac{1}{2} \cdot \mathbb{E}_{u \sim V} \left[ \max_{h \in \mathcal{H}_{|T}} \langle u, h \rangle \right].
\end{aligned}
$$

In the last step $u$ is sampled uniformly random from $V$. Notice that $u(x_i)$ equals $+1$ or $-1$ with probability $1/2$ independently for any $(x_i, y_i) \in S$. Therefore,

$$\Phi_{\text{discounted}}(V) = \frac{n}{2} \cdot \widehat{\Re}_T(\mathcal{H}_{|T})$$

where $\widehat{\Re}_T(\mathcal{H}_{|T})$ is the empirical Rademacher complexity which is defined as:

$$\widehat{\Re}_T(\mathcal{H}_{|T}) := \frac{1}{n} \mathop{\mathbb{E}}_{\sigma \sim \{+1,-1\}^n} \left[ \sup_{h \in \mathcal{H}_{|T}} \sum_{i=1}^{n} \sigma_i \cdot h(x_i) \right].$$

Finally, we recall the following lemma from Dudley (1967) and Haussler (1995).

**Lemma 11 (Implied by (Dudley, 1967) and (Haussler, 1995))**  *For a sample $T \in \mathcal{X}^n$ and finite VC dimension hypothesis class $\mathcal{H}$, we have*

$$\widehat{\Re}_T(\mathcal{H}_{|T}) \leq 31 \cdot \sqrt{\frac{\text{VC}(\mathcal{H})}{n}}.$$

Applying this lemma, we observe the following, and conclude the proof.

$$\Phi_{\text{discounted}}(V) = \frac{n}{2} \cdot \widehat{\Re}(\mathcal{H}_{|T}) \leq 16 \cdot \sqrt{n \cdot \text{VC}(\mathcal{H})}.$$

Lemma 11 is a well-known result derived using the chaining method introduced by Dudley (1967). Chaining is a technique for obtaining tight upper bounds on the expected supremum of a collection of random variables $\{X_t\}_{t \in T}$, denoted as $\mathbb{E}[\sup_{t \in T} X_t]$. This approach provides a sharper bound than the naive sum $\mathbb{E}[\sum_{t \in T} X_t]$, which can be loose when the variables are highly positively correlated. In such cases, it is beneficial to group together random variables that are nearly identical.

When calculating the Rademacher complexity $\widehat{\Re}_T(\mathcal{H}_{|T})$, the random variables are sums of the form $\sum_{i=1}^{n} h(x_i)\sigma_i$, where $h$ ranges over hypotheses forming a metric space under the Hamming distance. Hypotheses that are close in this metric space yield similar sum values. Dudley's integral (Dudley, 1967) allows us to bound the expected supremum in terms of the covering numbers (sphere coverings) of these hypotheses. Haussler (1995) provided an upper bound on the number of spheres that can cover a hypothesis class $\mathcal{H}$ with finite VC dimension, which directly translates to an upper bound on the covering numbers of $\mathcal{H}_{|T}$. Combining Dudley's chaining argument with Haussler's bound leads to Lemma 11.

For a formal proof of this lemma, see the lecture notes by Rebeschi (2022). For a self-contained introduction to chaining and its applications, we refer readers to Talagrand (1996).

## 4. Conclusion

This paper does a deep dive into the relationship between the transductive and PAC models of learning. We show that PAC learning is essentially no harder than transductive learning for most natural problems, by way of an efficient reduction. Though the converse appears challenging in general, we take the first step by showing that transductive binary classification is essentially no harder than its PAC counterpart. We speculate that further progress in this direction might come by extending our analysis of the agnostic OIG to the multiclass setting. In particular, characterizing

the subgraph of the agnostic OIG which maximizes discounted density, then relating its density to the PAC sample complexity, seems a natural and promising approach. Moreover, recent work by Hanneke et al. (2024) indicates that agnostic PAC learners can improve upon standard worst-case bounds when the hypothesis class error is small. We suspect that a transductive-to-PAC conversion that leverages small hypothesis class errors might yield PAC learners with performance guarantees that surpass those of traditional worst-case bounds.

## Acknowledgments

We thank the anonymous reviewers for their valuable comments and helpful feedback, including their suggestions for missing references. The first two authors received support from NSF Grants CCF-2009060 and CCF-2432219, while the third author was supported by NSF Grant CCF-2432219 and a Graduate Student Fellowship from the Viterbi School of Engineering.

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

## Appendix A. An alternative way to show the reduction in the realizable case

In this section, we revisit and extend the technique, originally presented by Aden-Ali et al. (2023a), for transforming transductive learners to PAC learners for realizable problems. This section serves as a warm-up for our main result: transforming agnostic transductive learners into agnostic PAC learners. That is, we prove the following theorem:

**Theorem 12** *(Equivalent to Aden-Ali et al. (2023a, Theorem 2.1)) For any realizable learning problem defined by sample space $\mathcal{X} \times \mathcal{Y}$ with a bounded pseudometric loss function $\ell(\cdot, \cdot) \in [0, 1]$ and a hypothesis class $\mathcal{H}$, we have*

$$m_{PAC,\mathcal{H}}(\epsilon, \delta) \leq m_{Trans,\mathcal{H}}(\epsilon/4) + \frac{3}{\epsilon} \cdot \log\left(\frac{2}{\delta}\right).$$

Our method of reduction, in simple terms, diverges from the approach taken by Aden-Ali et al. (2023a) in terms of problem framing. Their study centers on determining the optimal error bound given a sample of size $n$ by training transductive learners on subsets of the data. In contrast, our strategy explores the requisite additional data needed to transform transductive learners to PAC learners, by training on supersets of the original sample. For a more detailed comparison of the two approaches, see Appendix C.

---
**Algorithm 2** Reduction from PAC Learning to Transductive Learning in the realizable setting.

**Data:** $n$ i.i.d. samples $S$ from $\mathcal{D}$.
**Result:** Predictor $\widehat{h}$.
Let $A_{\text{Trans}}$ be a transductive learner with error rate $\epsilon(n)$.
Define $S_i := \{(x_1, y_1), \ldots (x_n, y_n), (x_{n+1}, y_{n+1}), \ldots (x_{n+i}, y_{n+i})\}$ for each $i \in [k]$
Let $h_{i-1}$ be the output predictor of $A_{\text{Trans}}(S_{i-1})$ for each $i \in [k]$.
Define $\widehat{h}$ as the predictor which returns the median of predictions $h_0(x), \ldots h_{k-1}(x)$.
**Output:** Predictor $\widehat{h}$.

---

We streamline the preceding analysis by adopting the following multiplicative variant of Azuma's inequality rather than formulating a specialized martingale concentration bound.

**Lemma 13 (Multiplicative Azuma's Inequality Kuszmaul and Qi (2021))** *Let $X_1, \dots X_n \in [0, c]$ be real-valued random variables for some $c > 0$. Suppose that $\mathbb{E}[X_i \mid X_1, \dots X_{i-1}] \leq a_i$ for all $i$. Let $\mu = \sum_{i=1}^n a_i$. Then for any $\delta > 0$,*

$$\mathbf{Pr}\left[\sum_i X_i \geq (1 + \delta) \cdot \mu\right] \leq \exp\left(-\frac{\delta^2 \cdot \mu}{(2 + \delta) \cdot c}\right).$$

Beyond simplifying the process, we have expanded the scope of our reduction to cover any supervised learning problem with bounded metric loss. This class of problems includes a wide array of supervised learning tasks, such as classification, partial concept classification, and bounded regression. For this extension, we have adapted the aggregation phase of the reduction to output the median of the prediction in the metric space, defining the median as the point that minimizes the total distance to all other predictions.

The following lemma is the key assertion that will allow us to finalize the proof of Theorem 12.

**Lemma 14** *Let $A_{Trans}$ be a transductive learner with transductive error rate $\epsilon(n)$. Then, for any $\delta \in (0, 1)$, and a sample $S \sim \mathcal{D}^{n+k}$, with probability $1 - \delta$ over the choices of $S$, predictors $h_{i-1} := A_{Trans}(S_{i-1})$ for $i \in [k]$ satisfy:*

$$\frac{1}{k} \sum_{i \in [k]} L_{\mathcal{D}}(h_{i-1}) \leq 4 \cdot \epsilon(n)$$

*where $k = \frac{3 \cdot \log(2/\delta)}{\epsilon(n)}$.*

Before we prove the lemma, let us show that how it implies Theorem 12.

**Proof** [Proof of Theorem 12] Fix any test point $(x, y)$, let $\widehat{h}$ be the predictor that outputs the median among predictions $h_0(x), \dots, h_{k-1}(x)$ which is the label $\widehat{y}$ that minimizes the total loss between $\widehat{y}$ and predictions $\{h_{i-1}(x)\}_{i \in [k]}$.

Thus,

$$\ell(y, \widehat{y}) \leq \min_{i \in [k]} \ell(h_{i-1}(x), \widehat{y}) + \ell(h_{i-1}(x), y) \qquad \text{(Triangle inequality)}$$

$$\leq \frac{1}{k} \sum_{i \in [k]} \ell(h_{i-1}(x), \widehat{y}) + \ell(h_{i-1}(x), y) \qquad (\min \leq \text{avg})$$

$$\leq 2 \cdot \frac{1}{k} \sum_{i \in [k]} \ell(h_{i-1}(x), y). \qquad (\widehat{y} \text{ is median})$$

Finally, taking expectation over samples $(x, y) \sim \mathcal{D}$ together with Lemma 14 concludes that distributional error of predictor $\widehat{h}$ is at most $8 \cdot \epsilon(n)$ with probability $1 - \delta$.

When $(\mathcal{Y}, \ell)$ constitutes a normed vector space, the application of Jensen's inequality ensures that the average of the predictors' outputs, $h_0(x), \dots h_{k-1}(x)$ yields at most $4 \cdot \epsilon(n)$ distributional error, thereby improves general metric space result. ∎

We now proceed to prove Lemma 14. To begin, let us define the empirical error of predictor $h_{i-1}$ for any $i \in [k]$ against the data point $(x_{n+i}, y_{n+i})$ as

$$d_i := \ell(h_{i-1}(x_{n+i}), y_{n+i}).$$

Given the sample set $S_{i-1}$, $d_i$ serves as an unbiased estimate of the distributional error associated with predictor $h_{i-1}$. Formally,

$$\mathbb{E}[d_i \mid S_{i-1}] = \mathbb{E}[\ell(h_{i-1}(x_{n+i}), y_{n+i}) \mid S_{i-1}] = L_{\mathcal{D}}(h_{i-1})$$

as we know that $(x_{n+i}, y_{n+i})$ is independently sampled from $\mathcal{D}$.

Next, we show that, with high probability, the total distributional error is at most a constant factor larger than the total empirical error. The key observation is that the cumulative differences of $\mathcal{L}_{\mathcal{D}}(h_{i-1}) - d_i$ form a martingale, and the sequence can be bounded using Azuma's multiplicative inequality.

**Claim 15 (Forward Martingale Bound)**

$$\mathbf{Pr}\left[\sum_{i=1}^{k} L_{\mathcal{D}}(h_{i-1}) - \sum_{i=1}^{k} d_i > 2 \cdot k \cdot \epsilon(n)\right] \leq \frac{\delta}{2}.$$

**Proof** Let $M_i = L_{\mathcal{D}, \mathcal{H}}(h_{i-1}) - d_i$. Then, as we observed above, the following conditional expectation evaluates to $\mathbb{E}[M_i \mid S_{i-1}] = L_{\mathcal{D}}(h_{i-1}) - \mathbb{E}[d_i \mid S_{i-1}] = 0$. Then, the random variables $M_i$ satisfy $M_i \in [-1, 1]$ and $\mathbb{E}[M_i \mid M_1, \ldots M_{i-1}] \leq \epsilon(n)$ where $\epsilon(n)$ is the error rate of the transductive learner $A_{\text{Trans}}$.

Invoking Lemma 13 gives us

$$\mathbf{Pr}\left[\sum_{i \in [k]} M_i \geq 2 \cdot k \cdot \epsilon(n)\right] < \exp\left(-\frac{k \cdot \epsilon(n)}{3}\right).$$

As we have $k = \frac{3 \cdot \log(2/\delta)}{\epsilon}$, the claim follows. ∎

Following that, we turn our attention to show that total empirical error is small with high probability. We observe that $d_i$ acts as an unbiased estimator for the transductive error of the learner $A_{\text{Trans}}$ given the unordered set $S_i$ as input. We then think of constructing our sequence of samples $S_0, \ldots, S_k$ "backwards" by first conditioning on the (unordered) set $S_k$, then iteratively peeling off one sample at a time — uniformly at random without replacement — to obtain $S_{k-1}, S_{k-2}, \ldots, S_0$. For $i = k, \ldots, 1$, observe that $(x_i, y_i)$ is a uniformly random element from $S_i$, and that $|S_i| \geq n$. The transductive error assumption implies that $L_{S_i}^{\text{Trans}}(A) \leq \epsilon(n)$. Finally, applying the multiplicative version of Azuma's inequality a second time, we obtain the following bound.

**Claim 16 (Backward Martingale Bound)**

$$\mathbf{Pr}\left[\sum_{i=1}^{k} d_i > 2 \cdot k \cdot \epsilon(n)\right] \leq \frac{\delta}{2}.$$

**Proof** Recall that we think of constructing our sequence of samples $S_0, \ldots, S_k$ "backwards" by first conditioning on the (unordered) set $S_k$, then iteratively peeling off one sample at a time — uniformly at random without replacement — to obtain $S_{k-1}, S_{k-2}, \ldots, S_0 = S$. For $i = k, \ldots, 1$,

observe that $(x_i, y_i)$ is a uniformly random element from $S_i$, and that $|S_i| \geq n$. The transductive error assumption implies that $L_{S_i}^{tr}(A) \leq \epsilon(n)$ and therefore

$$\mathbb{E}[d_i \mid S_i] \leq \epsilon.$$

Since $d_i$ is conditionally independent of $d_k, \ldots, d_{i+1}$ given $S_i$, it follows that

$$\mathbb{E}[d_i \mid d_k, \ldots, d_{i+1}] \leq \epsilon.$$

As we have $d_i \in [0, 1]$, invoking Lemma 13 once again gives us

$$\mathbf{Pr}\left[\sum_{i \in [k]} d_i \geq 2 \cdot k \cdot \epsilon(n)\right] \leq \exp\left(-\frac{k \cdot \epsilon(n)}{3}\right).$$

Since $k = \frac{3 \cdot \log(2/\delta)}{\epsilon}$, the claim follows. ∎

Combining two claims by using union bound, we conclude the proof of Lemma 14, from which Theorem 12 follows as previously described.

## Appendix B. Selecting a Hypothesis Using A Validation Set Gives a Low Error Hypothesis With High Probability

**Lemma 17** *Let $H := \{h_0, \ldots h_{k-1}\} \subseteq \mathcal{X}^{\mathcal{Y}}$ be a collection of predictors satisfying*

$$\mathbf{Pr}\left[\frac{1}{k} \cdot \sum_{i=0}^{k-1} L_{\mathcal{D},\mathcal{H}}^{Ag}(h_i) > \epsilon\right] \leq \delta$$

*for some $\epsilon, \delta \in [0, 1]$, and let $S_{val} := (\mathcal{X} \times \mathcal{Y})^t$ be a cross-validation set of size $t = \frac{\log(k/\delta)}{\epsilon^2}$, sampled i.i.d. from the distribution $\mathcal{D}$. Then, for the predictor $\widehat{h} := \mathrm{argmin}_{h_i \in H} L_{S_{val},\mathcal{H}}^{Ag}(h_i)$, which minimizes the agnostic loss on the cross-validation set, we have*

$$\mathbf{Pr}\left[L_{\mathcal{D},\mathcal{H}}^{Ag}(\widehat{h}) > 3\epsilon\right] \leq 2 \cdot \delta.$$

**Proof** Let $S_{val} := (x_1, y_1), \ldots, (x_t, y_t)$. Let $h^* \in H$ be the predictor with the minimum distributional loss among predictors in $H$. By assumption, with probability $1 - \delta$, the average distributional loss of predictors $h \in H$ is at most $\epsilon$. From now on, we condition on this event and assume that the average agnostic distributional error is at most $\epsilon$. Therefore, optimal $h^*$ guarantees that $L_{D,\mathcal{H}}^{Ag}(h^*) \leq \epsilon$. For $h^*$, define $X_i := \ell(h^*(x_i), y_i)$ for $i \in [t]$.

Using Hoeffding's inequality and the fact that $X_i \leq 1$ for each $i \in [t]$, we observe that

$$\mathbf{Pr}\left[L_{S,\mathcal{H}}^{Ag}(h^*) \geq \epsilon + \sqrt{\frac{\log(1/\delta)}{2 \cdot t}}\right] = \mathbf{Pr}\left[\frac{1}{t}\sum_{i=1}^{t} X_i \geq \epsilon + \sqrt{\frac{\log(1/\delta)}{2 \cdot t}}\right] \leq \delta. \qquad (5)$$

On the other hand, let $h \in H$ be an arbitrary predictor with $L_{D,\mathcal{H}}^{Ag}(h) \geq \epsilon + 2 \cdot \sqrt{\frac{\log(k/\delta)}{2 \cdot t}}$. Using Höeffding's bound again, we observe that

$$\mathbf{Pr}\left[L_{S,\mathcal{H}}^{Ag}(h) \leq \epsilon + \sqrt{\frac{\log(k/\delta)}{2 \cdot t}}\right] \leq \frac{\delta}{k}.$$

By the union bound, we can conclude that with probability at least $1 - \delta$, there is no predictor $h \in H$ with $L_{S,\mathcal{H}}^{\text{Ag}}(h) \leq \epsilon + \sqrt{\frac{\log(k/\delta)}{2 \cdot t}}$ and $L_{D,\mathcal{H}}^{\text{Ag}}(h) \geq \epsilon + 2 \cdot \sqrt{\frac{\log(k/\delta)}{2 \cdot t}}$. By (5), we know that with probability $1 - \delta$, there exists a predictor whose agnostic empirical loss is at most $\epsilon + \sqrt{\frac{\log(k/\delta)}{2 \cdot t}}$. Therefore, we conclude that $\widehat{h}$, the empirically optimal predictor, attains at most $\epsilon + 2 \cdot \sqrt{\frac{\log(k/\delta)}{2 \cdot t}}$ agnostic distributional error.

Finally, we release the condition on the average distributional error being at most $\epsilon$ and conclude that

$$\mathbf{Pr}\left[ L_{\mathcal{D},\mathcal{H}}^{\text{Ag}}(\widehat{h}) \geq \epsilon + 2 \cdot \sqrt{\frac{\log(k/\delta)}{2 \cdot t}} \right] \leq 3 \cdot \delta.$$

The proof is complete as we set $t = \frac{\log(k/\delta)}{\epsilon^2}$. ∎

## Appendix C. Connections to the results of Aden-Ali et al.

In this section, we explore the connection between our work and that of Aden-Ali et al. (2023a). Both papers present techniques for converting transductive learners to PAC learners in the realizable setting, but they make different assumptions about the properties of the underlying learners. Aden-Ali et al. (2023a) train multiple transductive learners by taking subsamples of the given sample set and aggregating their information to obtain a PAC learner. Their work explicitly defines this aggregation step for binary classification, partial concept classification, and bounded loss regression problems. In contrast, our approach involves training multiple learners by providing additional data to each and aggregating their outputs to form a PAC learner. Our aggregation method is more generic and applicable to any learning problem with bounded pseudometric loss functions.

Both our paper and the paper of Aden-Ali et al. (2023a) make assumptions on the worst case performance of a learner $A$ based on the size of the input sample. Intuitively, this paper assumes that when a learner is given larger samples, its expected error on those samples is non-increasing. That is,

$$\epsilon_{\text{Trans}}(n) \leq \epsilon_{\text{Trans}}(n - 1). \tag{A1}$$

In contrast, the authors of Aden-Ali et al. (2023a) assume that given smaller samples, the expected error of a learner does not increase more than linearly. More formally,

$$n\epsilon_{\text{Trans}}(n) \geq (n - 1)\epsilon_{\text{Trans}}(n - 1). \tag{A2}$$

As you may notice, these two assumptions are incomparable, or equivalently, one does not imply the other.

In this section, we will first show that A1 is without loss of generality true for transductive learners. We then show that assuming both (A1) and (A2) hold for transductive learners, the two results are equivalent up to constant factors.

### C.1. The monotonicity and symmetry assumptions in Section 2 are true without loss of generality

**Lemma 18** *If a learner $\widetilde{A}$ achieves (agnostic) transductive error at most $\widetilde{\epsilon}(n)$, then there exists a randomized learner $A$ with transductive error rate of at most $\epsilon(n)$ such that $\epsilon(n + 1) \leq \widetilde{\epsilon}(n)$.*

**Proof** For any sample $S \in (\mathcal{X}, \mathcal{Y})^{n+1}$ provided for transductive learning task. When a transductive learner $A$ is tested against data point $(x_i, y_i)$, $A$ subsamples a set $S' \subseteq S_{-i}$ of size $n - 1$ uniformly at random and returns the label $A'(S')(x_i)$. We can upper-bound transductive error of $A$ as follows.

$$
\begin{aligned}
L_S^{\text{Trans}}(A) &= \underset{i \in [n]}{\mathbb{E}} \left[ \underset{j \in [n] \backslash \{i\}}{\mathbb{E}} \left[ \ell(A'(S_{-i,-j})(x_i), y_i) \right] \right] \\
&= \underset{i \in [n]}{\mathbb{E}} [L_{S_{-i}}^{\text{Trans}}(A')] \\
&\leq \underset{i \in [n]}{\mathbb{E}} [\widetilde{\epsilon}(n)] \\
&= \widetilde{\epsilon}(n).
\end{aligned}
$$

$\blacksquare$

Notice that the proof holds even for agnostic transductive learning problem when we replace the transductive error function with the agnostic transductive error function.

Moreover, for transductive learners with VC dimension $d$, we know that $\epsilon(n) = O\left(\frac{d}{n}\right)$ and $\epsilon_{\text{Ag}}(n) = O\left(\sqrt{\frac{d}{n}}\right)$. Therefore, Assumption A1 holds for these learners as well.

Let us say that we have a learner $A$ which is not symmetric and attains transductive error guarantee $L_S^{\text{Trans}}(A) \leq \epsilon_{\text{Trans}}$. We note that we can construct a learner $A'$, where $A'(S) = A(\pi(S))$ where $\pi$ is a permutation chosen as a function of the set of (unlabeled) datapoints in $S$. Note that because $A(S) \leq \epsilon_{\text{Trans}}$ regardless of the order of $S$, we see that $A(\pi(S)) \leq \epsilon_{\text{Trans}}$ as well. In addition, we can see that $A'$ is symmetric over input samples, as $\pi$ sorts the sample consistently. Therefore, we can conclude that if a learner $A$ exists attaining transductive error rate $\epsilon_{\text{Trans}}$, then a symmetric learner $A'$ exists which also attains transductive error $\epsilon_{\text{Trans}}$.

