# OpenReview forum: "Is Transductive Learning Equivalent to PAC Learning?"
_algorithmiclearningtheory.org/ALT/2025/Conference — ALT 2025_

### Official Review · Reviewer_KgJr · 2024-10-17

**Rating:** 7
**Confidence:** 5

**Review:**

The paper studies the sample complexities of agnostic PAC and transductive learning. These two settings are closely related in the realizable setting where the labels obtained from the data generating distribution, $D$, are consistent with some hypothesis from a function class $H$: optimal predictors for the transductive setting also achieve optimal in-expectation guarantees for the PAC setting, with recent results establishing that these guarantees can be adapted to yield optimal high-probability PAC learners. Hence, the sample complexities of the PAC and transductive settings are equivalent for the realizable setting. On the other hand, the agnostic setting, where the distribution may be arbitrary and the goal is to compete with the best predictor from $H$, is far less clear: 1) It is clear that in terms of in-expectation guarantees, a predictor with $\epsilon$ excess error in the transductive setting implies one with $\epsilon$ excess error in the PAC setting, and 2) however, going in the other direction, the best-known result shows that a predictor with excess error $\epsilon$ can only yield a predictor with polynomially worse (in $1 / \epsilon$) excess error in the transductive setting. This paper aims to address that gap with two contributions: 1) Firstly, they extend recent work which establishes that transductive predictors can yield high-probability PAC predictors with the same error (plus a term dependent on the failure probability) in the realizable setting to the agnostic setting and 2) in the specific setting of binary classification, they precisely characterize the best possible achievable excess error in the transductive setting and show that this implies that the sample complexities of PAC and transductive learning are equivalent for agnostic binary classification.

Technically, the first result follows from a modification of a similar technique used in the realizable setting. The prior approach, when naively adapted in this setting, suffers from a multiplicatively worse dependence on the error of the best hypothesis from the class. In this paper, they show that maintaining a small hold-out set is sufficient to avoid this dependence. The main technical contribution of the paper is the second result: they show that the excess transductive risk is upper bounded by the Rademacher complexity (scaled by $\sqrt{n}$) of the function class. This is a surprising finding and relies on a clever symmetrization argument. It establishes that the optimal transductive and PAC sample complexities essentially coincide for agnostic binary classification.

Overall, the findings of the paper are interesting and non-trivial and suggest exciting directions for future research. Hence, I recommend acceptance. I do have two comments for the authors: 1) It appears that Algorithm 1 requires $\delta$ to be known in advance? Is this strictly necessary since the prior approach does not require this? and 2) There are recent results that show that the sample complexity of agnostic PAC learning may be improved when the error of the best hypothesis is small (https://arxiv.org/pdf/2407.19777). Perhaps the authors could comment on whether such a result is possible for their approach as well?

**Paper Award:**

No

---

### Official Review · Reviewer_FtZN · 2024-11-07
**Review of "Is Transductive Learning Equivalent to PAC Learning?"**

**Rating:** 7
**Confidence:** 3

**Review:**

The paper considers the theoretical problem of essential equivalence between Transductive Learning and Probably Approximately Correct (PAC) Learning in supervised learning problems. Here, two learning models are essentially equivalent if and only if their sample complexity only differs by an additive polynomial in terms of the error rate $\epsilon$ (in both transductive and PAC learning) and the tail probability $\delta$ (in PAC learning). The authors establish a series of results on essential equivalence between these two models, with a focus on agnostic setting.

Key contributions of the paper:

1. Realizable learning setting: This paper extends the prior results of Aden-Ali et al. (2023a) on realizable setting to cover the entire class of (pseudo) metric losses and provides a unified proof.

2. Agnostic learning setting: The authors show that transductive learning and PAC learning are essentially equivalent for agnostic setting with bounded losses and in binary classification, and further conjecture that this is true for most natural label spaces and loss functions.

Overall, I think this is a good paper. This paper is well written and the main results are presented in a way that is easy to understand. By extending existing results to agnostic settings, the authors address an area where prior results were limited, especially in binary classification. Their technical approach, including symmetrization arguments and the OIG framework, is also well-presented and easy to follow.

**Paper Award:**

No

---

### Official Review · Reviewer_EXSR · 2024-11-14
**They give a confidence-amplification technique to convert in-expectation agnostic transductive guarantees to high-probability agnostic PAC guarantees.  This seems useful.**

**Rating:** 7
**Confidence:** 5

**Review:**

The paper studies the relation between (single test point) transductive learning (which concerns in-expectation performance guarantees for permutations of n+1 fixed examples) and PAC learning (which concern high-probability performance guarantees from i.i.d. samples).  Such relations are well known in the realizable case (with a sharp transductive-to-PAC transformation appearing in the recent work of Aden-Ali, Cherapanamjeri, Shetty, and Zhivotovskiy, 2023).  The main contribution of this work is extending this relation to the agnostic setting: effectively providing a confidence-amplification tool for converting in-expectation epsilon excess error guarantees into high-probability 1-delta guarantees of excess error O(epsilon), at only the expense of an additive (1/epsilon^2)log(1/epsilon*delta).  The result holds in an abstract setting, with general label spaces and bounded loss functions.

I believe this result is interesting, and will be a nice addition to the learning theory toolbox.

The actual technique is essentially similar to that of Aden-Ali et al. from the realizable case, based on applying the in-expectation learner to a sequence of prefixes of the data, and constructing a final predictor based on the resulting set of hypotheses (e.g., the simplest strategy for this is to construct a randomized predictor that uniformly samples from these hypotheses at test time, though they also discuss constructing non-randomized predictors which select one hypothesis via a validation set).  The analysis, though rooted in the arguments of Aden-Ali et al., is more nuanced than the realizable case, in several ways that make this a non-trivial extension (for instance, being careful not to incur multiplicative increases in error rate, and accounting for different competitor functions for the different prefixes).

A second contribution is an analysis of transductive agnostic learning for the special case of binary classification.  In this case, they argue that a discounted edge density of the agnostic one-inclusion graph is bounded by O(\sqrt{n*VC}), via a connection to the empirical Rademacher complexity.  This translates into an expected excess error O(\sqrt{VC/n}) in the agnostic transductive setting, which matches the known optimal guarantee for the i.i.d. setting.

Altogether, I believe the results would be nice to have in the published literature, as they should be useful for future work on agnostic PAC learning in certain settings where there is ongoing development (e.g., the multiclass setting).

Comments for the authors:

- The paper is citing Asilis, Devic, Dughmi, Sharan, and Teng (2024) as proposing the agnostic one-inclusion graph predictor.  But actually this is a well-known idea introduced much earlier in the work of Phil Long (1999) "The Complexity of Learning According to Two Models of a Drifting Environment".  (indeed, among some of us in the community, orientations achieving O(\sqrt{VC/n}) excess risk have been known for some time, though I believe your paper will be the first appearance of this in a publication)

- It is mentioned in a couple places (e.g., page 3) that the transductive setting was first proposed and studied by Haussler, Littlestone, and Warmuth (1994).  But in fact it originates in the work of Vapnik and Chervonenkis who studied it in substantial detail (in both realizable and agnostic settings, both in leave-one-out type guarantees and high-probability variants); e.g., see their 1974 book, or Vapnik's 1982 or 1998 books.

- An important natural comparison to include is techniques for online-to-batch conversion, which are essentially based on similar martingale-type concentration arguments for applying a learner to prefixes.
See for instance the literature rooted in the article "On the Generalization Ability of On-Line Learning Algorithms" by Cesa-Bianchi, Conconi, and Gentile (2004).

- In Theorem 2, is the epsilon(n) in the log a typo?  i.e., should it just be epsilon?

**Paper Award:**

No

---

### Author Rebuttal · Authors · 2024-11-25

We sincerely thank all the reviewers for their valuable and constructive feedback.

We thank the first reviewer for pointing out the origins of these ideas. We will revise our discussion of related work accordingly. We will also expand on the connection with online-to-batch conversion. The first reviewer is also correct that epsilon(n) is a typo.

Regarding the third reviewer's first question, dependence on $\delta$ appears unavoidable with our current approach, though we cannot rule out different approaches which remove such dependence. For the second question, we first recall that the agnostic setting uses the additive form of Azuma's inequality, while the realizable setting employs the multiplicative version due to small absolute errors in the realizable case. We suspect that one can derive a martingale concentration inequality which interpolates between the two, and in doing so leads to better bounds when the error of the best hypothesis is small. We thank the reviewer for pointing out these potential extensions, and we will mention them as ideas for future work in the next version of the paper.

---

### Meta-Review · Area_Chair_NsUM · 2024-11-28

**Recommendation:** Accept
**Confidence:** 4

**Metareview:**

The paper studies the relation between (single test point) transductive learning (which concerns in-expectation performance guarantees for permutations of n+1 fixed examples) and PAC learning (which concern high-probability performance guarantees from i.i.d. samples). The referees unanimously recommend acceptance.

**Paper Award:**

No